# Early intervention services for non-psychotic mental health disorders: a scoping review protocol

Katie Richards ,[1] Amelia Austin ,[1] Karina Allen,[1,2,3] Ulrike Schmidt[1,2]

¹Psychological Medicine, King's College London, London, UK
²Eating Disorder Outpatients Service, South London and Maudsley Mental Health NHS Trust, London, UK
³School of Psychological Science, The University of Western Australia, Perth, Western Australia, Australia

**Correspondence to**
Katie Richards;
katie.richards@kcl.ac.uk

## ABSTRACT

**Introduction** Worldwide mental health disorders are associated with a considerable amount of human suffering, disability and mortality. Yet, the provision of rapid evidence-based care to mitigate the human and economic costs of these disorders is limited. The greatest progress in developing and delivering early intervention services has occurred within psychosis. There is now growing support for and calls to extend such approaches to other diagnostic groups. The aim of this scoping review is to systematically map the emerging literature on early intervention services for non-psychotic mental health disorders, with a focus on outlining how services are structured, implemented and scaled.

**Methods and analysis** The protocol was developed using the guidance for scoping reviews in the Joanna Briggs Institute manual and the Preferred Reporting Items for Systematic Reviews and Meta-Analyses extension for scoping reviews checklist. A systematic search for published and unpublished literature will be conducted using the following databases: (1) MEDLINE, (2) PsycINFO, (3) HMIC, (4) EMBASE and (5) ProQuest. To be included, documents must describe and/or evaluate an early intervention service for adolescents or adults with a non-psychotic mental health disorder. There will be no restrictions on publication type, study design and date. Title and abstract, and full-text screening will be completed by one reviewer, with a proportion of articles screened in duplicate. Data analysis will primarily involve a qualitatively summary of the early intervention literature, the characteristics of early intervention services and key findings relating to their evaluation and implementation.

**Ethics and dissemination** The synthesis of published and unpublished articles will not require ethical approval. The results of this scoping review will be published in a peer-reviewed journal and disseminated via social media, conference presentations and other knowledge translation activities.

## INTRODUCTION

Early intervention is widely perceived as beneficial in medicine and refers to the early detection and initiation of stage-specific treatment.[1] Proactive treatments matched to the stage of illness can limit or even avert unfavourable outcomes, reducing the need for costly and more invasive treatments in the future.[2 3] Despite such promise, early

## Strengths and limitations of this study

► This scoping review will provide a comprehensive overview of both published and unpublished literature for the emerging research field of early intervention services for non-psychotic mental health disorders.

► The review will be conducted according to the standardised methodology outlined in the Joanna Briggs Institute manual and using the Preferred Reporting Items for Systematic Reviews and Meta-Analyses checklist for scoping reviews.

► Part of the screening and charting process will be completed in duplicate to ensure reliability of these methods.

► Only articles written in English, German, French and Spanish will be included, the review may, therefore, be biassed.

intervention approaches have been slow to gain momentum in mental health.[4 5] Mental illnesses are a major contributor to mortality and disability worldwide, particularly for young people.[6–8] The typical age of onset for mental disorders is adolescence and early adulthood (12–30 years), a period of marked social, psychological and biological change.[9 10] A delay in or lack of access to effective treatments during this time could disrupt key developmental milestones and have long-lasting effects on health, social and occupational trajectories.[11]

Service provision does not match the topography of onset or burden of disease associated with mental disorders, even in relatively well-developed health systems.[12] Globally, access to evidence-based care is poor, and even for those that do access it, this is often after lengthy delays.[13–15] The duration of untreated illness (DUI), defined as the period between the onset of psychiatric disorder and the initiation of treatment, ranges from 1 to 2 years for psychosis to 10 years for obsessive–compulsive disorder (OCD).[16–19] Over time, mental disorders can become more entrenched through functional deterioration, neuroadaptation

and habitual behaviour patterns.[20–23] Indeed, a longer DUI is associated with worse symptomatic and functional outcomes, and a lower treatment response across diagnostic groups.[19 24–27] More worryingly, young people, the group at highest risk for psychiatric difficulties, tend to have the worst access to timely care.[13 18 28–30]

Together, such findings provide a compelling case for establishing early intervention services that match the developmental needs and symptomatic profile of individuals with recent-onset mental disorders.[4 14] The greatest strides in early intervention have been made within psychosis. Over the past 30 years, early intervention for psychosis (EIP) has gained tremendous support from researchers and healthcare professionals worldwide.[14] EIP services have two fundamental aims: to reduce the duration of untreated psychosis, and to provide evidence-based, stage-specific treatment.[31] EIP services use a clinical staging approach to map the extent of illness progression from early presymptomatic risk to severe and enduring, enabling a prevention orientated framework that matches the intensity of treatment to the level of need.[32 33] A comprehensive body of high-quality research shows that compared with standard care, multicomponent EIP services are associated with a reduction in symptom severity, relapse rates and hospitalisation risk, as well as improved global functioning and quality of life.[34] Moreover, consistent evidence suggests that EIP services are a cost-effective alternative to standard care.[35] There has been a recent surge in papers calling for early intervention approaches to be broadened to other diagnostic groups, including major depression,[36] OCD,[22] eating disorders[37] and bipolar disorder.[38] Preliminary evidence from services for recent-onset eating and mood disorders demonstrate significant improvements in symptoms, reduced hospital (re)admissions, and most importantly, high levels of patient satisfaction.[39–42]

The utility of focusing exclusively on discrete diagnostic categories in the delivery of early intervention specifically, and mental healthcare more generally has, however, been questioned.[32 43] The early stages of mental disorder are often characterised by fluctuating patterns of specific and non-specific subthreshold symptoms, diagnostic instability and comorbidity.[44 45] A single-disorder focus could result in these earlier presentations of illness being excluded.[46] A transdiagnostic approach, consistent with evidence for pluripotent models of clinical staging, has been put forward as a necessary solution to address this problem.[32 43 47 48] The recognition of the need to broaden the early intervention paradigm has led to the development of several integrated youth mental health hubs.[49 50] These hubs act as entry-level services for young people irrespective of diagnosis, and typically provide a comprehensive package of low-intensity mental, physical and social care support in community settings. Young people tend to rate these services positively and between 52% and 68% experience improvements in symptoms and functioning. However, a proportion of individuals with more severe symptoms do not seem to benefit from these services and rigorous outcome research for youth hubs is limited.[50 51]

Although the role of early intervention in reducing distress and functional impairment seems obvious, the evidence-base for these services is incomplete and much more work needs to be done.[14 22] There is limited prospective evidence evaluating the utility of these services for non-psychotic disorders, it is unclear to what extent the findings from psychosis would translate to other diagnostic groups. There is also a lack of research evaluating the feasibility or the implementation processes of services in clinical settings.[51] Moreover, even within psychosis, further research is needed to determine how long EIP services should be provided, whether it is the reduction in DUI or other components of EIP services that account for the improved outcomes, and whether outcomes would be similar with other service structures and models.[52 53] An ever-growing population, accompanied by reducing health budgets, creates an environment where only services that demonstrate effectiveness, economic viability and sustainability receive funding.[54] It is, therefore, imperative to develop a rigorous evidence base to refine, adapt and evaluate early intervention services for non-psychotic disorders, with a particular focus on identifying the "active ingredients" of such services and the most effective methods for widespread scaling and implementation.

The primary objective of this review is to provide a baseline characterisation of the differing ways in which early intervention services are structured and implemented for non-psychotic mental health disorders. The emerging literature for non-psychotic disorders is heterogeneous and dispersed, with distinct streams of research developing in disciplinary silos. The aim of this review is to draw together these streams to facilitate collaboration and cross-disciplinary learning and discourse. By synthesising the field and highlighting commonalities and differences, we hope that a broad set of common principles for early intervention services will emerge. This review, in conjunction with reviews in psychosis, will help set the stage for a more unified approach to expanding and refining early intervention services for psychiatric disorders. Here, we focus exclusively on disorders that tend to emerge in adolescence and adulthood rather than in childhood. Neurodevelopmental disorders typically use a very different approach to early intervention than adolescent-onset and adult-onset disorders (eg, intervening in infancy).[55] A scoping review methodology was selected for this review as early intervention is an emerging, dispersed and heterogeneous research area and is therefore not amenable to the narrower aims of a traditional systematic review.[56 57] Given that this is a relatively new research area, we sought to map all the available evidence within this field rather than only the best available evidence (eg, randomised controlled trials).[58]

**Table 1** MEDLINE search strategy

| | Query | Results |
|---|---|---|
| #1 | exp Early Medical Intervention [MeSH term)/ or (early intervention* or early-intervention*).tw | 19 623 |
| #2 | exp Mood Disorders [MeSH term)/ or Bipolar Disorders [MeSH term)/ or (mood disorder* or affective disorder* or depressi* or dysthymi* or bipolar*).tw | 453 041 |
| #3 | #1 AND #2 | 1616 |
| #4 | exp Anxiety Disorders [MeSH term)/ or (anxiety disorder* or neurotic disorder* or agoraphobi* or obsessive-compulsive disorder* or OCD or panic disorder* or phobic disorder* or post-traumatic stress disorder* or post traumatic stress disorder* or PTSD or generalised anxiety disorder* or social phobia).tw | 119 604 |
| #5 | #1 AND #4 | 560 |
| #6 | exp "Feeding and Eating Disorders" [MeSH term)/ or (eating disorder* or anorexi* or bulimi* or binge-eating* or binge eating* or (eating disorder not otherwise specified) or EDNOS or (other specified feeding or eating disorder) or OSFED).tw | 56 480 |
| #7 | #1 AND #6 | 199 |
| #8 | exp Substance-Related Disorders [MeSH term)/ or exp "Disruptive, Impulse Control, and Conduct Disorders" [MeSH term)/ or (((substance-related or alcohol or opioid or morphine or marijuana or heroin or cocaine or amphetamine or cannabis) adj1 (disorder* or illness* or dependence or abuse or misuse)) or (impulse control disorder*) or conduct disorder* or fire setting behaviour* or gambling or trichotillomania).tw | 295 108 |
| #9 | #1 AND #8 | 924 |
| #10 | exp Somatoform Disorders [MeSH term)/ or (somatoform or somatoform disorder* or somati#ation or body dysmorphi* or conversion disorder* or hypochondri*).tw | 25 487 |
| #11 | #1 AND #10 | 38 |
| #12 | exp Personality Disorders [MeSH terms)/ or (personality disorder* or antisocial personality disorder* or anti-social personality disorder* or borderline personality disorder* or emotionally unstable personality disorder* or obsessive-compulsive personality disorder* or dependent personality disorder* or histrionic personality disorder* or narcissistic personality disorder* or avoidant personality disorder* or paranoid personality disorder* or schizoid personality disorder* OR schizotypal personality disorder*).tw | 47 019 |
| #13 | #1 AND #12 | 208 |

## RESEARCH QUESTIONS

1. What is the extent, range and nature of the literature on early intervention services for adolescents and adults with non-psychotic mental health disorders?
2. What are the characteristics of early intervention services and care pathways?
   a. Are there any similarities and/or differences across early intervention services provided for each diagnosis and transdiagnostically?
3. Are there any factors that influence the implementation of early intervention services (ie, barriers and facilitators to implementation)?
4. Do early intervention services reduce DUI, improve the course and outcome of mental disorders or minimise the disruption to psychosocial development and function?

## METHODS AND ANALYSIS

The Preferred Reporting Items for Systematic Reviews and Meta-Analyses extension for scoping reviews (PRISMA-ScR) checklist,[57] and the scoping review framework outlined in the Joanna Briggs Institute (JBI) Reviewer's Manual[59] were used to guide the development of this protocol.

### Eligibility criteria

Documents will be included if they: (1) Describe and/or evaluate an early intervention service for non-psychotic mental health disorders (concept) based in any type of healthcare facility (ie, hospitals, day services and community settings) and in any geographical area (context). Here, early intervention refers to a structured programme of care delivered by a stand-alone team or teams integrated into mental health services that provide treatment for individuals with recent-onset subthreshold or threshold disorders. The level of care can vary from low-intensity techniques of signposting, psychoeducation and self-help resources all the way through to specialised multidisciplinary teams and complex high intensity interventions; (2) Describe and/or evaluate an early intervention service for adolescents (≥10–17 years) or adults (>18 years) with a recent-onset subthreshold or threshold mood disorder, anxiety disorder, eating disorder, personality disorder, impulse control or substance use disorder, and/or somatoform disorder (types of participants).

**Table 2** Draft data charting form

| Data item | Description of item |
|---|---|
| **Document details** | |
| Type of document | The type of document can include but will not be limited to published or unpublish primary research, any type of review, protocols, theoretical paper, guidelines, opinion pieces, editorials and expert consensus papers. |
| Author(s) | List of authors |
| Year of publication | Year of publication |
| Title | Title of document |
| Journal | The title of the scientific journal (for published documents only) |
| Country of origin | Country where the document originates |
| Aim/purpose of document | Summary of the aim/purpose of the document |
| Study design | For published or unpublished research papers, the design of the study as reported in the paper. Includes but is not limited to randomised controlled trials, pre–post design, historical controlled trial, prospective or retrospective cohort studies, cross-sectional and case series/study. |
| Study methodology | The methodological framework: qualitative, quantitative or mixed methods. |
| **Characteristics of early intervention service** | |
| Name of service | The name of the early intervention service/programme. |
| Year established | The year the early intervention service was established. |
| Location | The country and region in which the early intervention service was implemented. |
| Population | The population for which the service was designed for. This item will include details such as age, diagnosis, duration of illness and illness severity. |
| Setting | The physical setting in which the early intervention service is based. This includes but is not limited to community centres, primary care, outpatient clinics and inpatient wards. Early intervention services can occupy more than one of these settings. |
| Service providers | A description of who provides the service and their role, includes but is not limited to social workers, youth workers, peer support workers, nurses, clinical or counselling psychologists and psychiatrists. |
| Service structure/ process | A description of the service structure and administrative processes includes but is not limited to 'service within a service' models, stand-alone multidisciplinary team models, 'hub' and 'spoke' models, and process variables such as specific wait time targets. |
| Access to service | Methods for accessing the early intervention service, includes but is not limited to active engagement and outreach through schools, colleges and youth clubs, referral from primary care, self-referral and drop-in. |
| Services and interventions | A description of the types of services and interventions provided, includes but is not limited to psychoeducation, online self-help and self-management support, psychological therapies (eg, CBT, brief therapy), sexual health and family planning, health promotion, social services, peer support, and crisis intervention and management. |
| Clinical staging | Whether a clinical staging approach was used to inform the design, evaluation or implementation of the service. |
| **Outcome Research** | |
| Participants | Details related to the participants included in the study. This will include information related to sample size, diagnosis, age, sex and inclusion/exclusion criteria. |
| Comparator data or standard care | Description of comparator data or the care provided to a control group. |
| Outcomes and time points | Description of the qualitative and quantitative outcomes and the time points of data collection. This will include standardised clinical assessments, and self-report measures as well as implementation outcomes, such as measures of acceptability, feasibility, adoption, fidelity and sustainment. |
| Key results/findings | An outline of the key results and findings reported in the document. This includes quantitative outcomes such as changes in symptoms, engagement and patient satisfaction, as well as qualitative outcomes, such as, descriptions of barriers and facilitators to implementation. |

CBT, cognitive–behavioural therapy.

| Table 3 | Summary of reach, effectiveness, adoption, implementation and maintenance framework criteria |
|---|---|
| Reach (participant representativeness) | The representativeness of individuals enrolled in the study to the characteristics of the intended population. <br> 1=Limited generalisability: highly selected subsample that is not typical of the intended population, high number of exclusionary criteria, and/or a recruitment strategy that is likely to result in a biassed sample. <br> 2=Moderately generalisable: participants match intended population on key characteristics (eg, sex/gender, diagnosis, age), but are still a selected subsample due to exclusion criteria and recruitment strategies. <br> 3=Generalisable: participants are typical of the intended population, limited or no exclusion criteria and/or recruitment strategies is not selective and are unlikely to result in a biassed sample. |
| Effectiveness (outcome representativeness) | Measured outcomes are important and meaningful to all stakeholders involved, including potential negative effects, quality of life and economic outcomes. <br> 1=Limited generalisability: primary outcomes restricted to an estimate of the overall effect of the intervention on a single metric of health, limited attention to process outcomes, quality of life, patient and staff satisfaction, patient engagement, unintended harms, or functional rehabilitation. <br> 2=Moderate generalisability: primary outcomes focus on overall effect of intervention on health, some inclusion of measures that are meaningful to stakeholders or process outcomes. <br> 3=Generalisable outcomes: primary outcomes include mix of impact of intervention on health and outcomes that are meaningful to patients and other stakeholders (including qualitative evaluations), explicit discussion around prevention of harms to participants, process outcomes, patient engagement, acceptability and satisfaction. |
| Adoption (setting representativeness) | The representativeness of settings and the individuals within those settings who deliver the programme. <br> 1=Limited generalisability: highly selected settings and staff and/or only includes 'best' sites and staff, that is, well-resourced, credentialed or seasoned interventionists, many exclusion criteria; or limited information to determine context of study or intervention. <br> 2=Moderate generalisability: intervention tested in contexts outside of 'best' sites and staff, but adoption is still limited to selected settings that are well resourced with some expertise in intervention trials. <br> 3=Generalisable: sites and staff are randomly selected, few or no exclusion criteria and/or trialled in diverse settings. |
| Implementation (fidelity/adaptation, and cost/feasibility) | Fidelity to the intervention and adaptations made to intervention during study/programme. <br> 1=Limited information on the implementation: no details on adaptation to local context, no details related to core element of interventions, or an evaluation of the consistency of implementation across settings staff, and patients. <br> 2=Moderate reporting of fidelity/adaptations: core elements described but details missing, or fidelity was monitored but no details on measurement tools. <br> 3=Detailed report of modifications made, adaptations to local context, and rationale for modification, an outline of core elements and evaluation of the fidelity to core elements of the model. <br><br> The cost of the intervention in terms of time and money. <br> 1=No details on time, cost and resources, no efforts to contain costs, and use of state-of-the-art resources and procedures such that costs of intervention are likely to be high. <br> 2=Details on time, cost and resources is still limited but more than for a rating of 1. The intervention has minimal impact on time, cost and resources. <br> 3=Explicit efforts to contain costs and to make the intervention feasible in low resource settings. |
| Maintenance (sustainment) | The extent to which an intervention becomes institutionalised or part of the routine organisational practices and policies and the extent to which behaviour is sustained for more than 6 months. <br> 1=Limited sustainability efforts or details of such efforts: no report of efforts to continue an intervention after the completion of study, or no reports of continued use. <br> 2=Moderate sustainment: limited discussion regarding the sustainability of an intervention, some evidence of continued use. <br> 3=Sustainment: long-term outcomes reported, explicit plans for handing off intervention to setting/sites, details of methods to encourage sustainable implementation or embedding within routine organisational practices and policies or evidence of sustained use for 6 months or more. |

Transdiagnostic early intervention services and early intervention services for comorbid/concurrent disorders will be included provided that at least one of the diagnoses is listed in the previous sentence; (3) Mixed child and adolescent services will be included, where feasible, only information relevant for the adolescent portion of the services will be charted and (4) All document types and study designs are eligible for inclusion: randomised

controlled trials, non-randomised studies, observational studies, qualitative studies, reviews, ongoing trials, protocols, theoretical papers, grey literature, editorials, opinions pieces and expert consensus statements (types of studies).

Documents will be excluded if they: (1) Describe a primary prevention programme based in educational establishments, high-risk groups (eg, athletes) or in the general population, (2) Describe a parent-only intervention, (3) Describe a specific intervention (eg, type of cognitive–behavioural therapy) that is not attached to a service and (4) Primarily or only focus on early intervention for a physiological or medical condition, schizophrenia spectrum and other psychotic disorders and/or neurodevelopmental disorders.

### Search strategy

A comprehensive literature search will be conducted from inception on PsycINFO, MEDLINE, EMBASE and HMIC. ProQuest databases will also be searched for grey literature (ie, conference papers and proceedings, theses, government publications). The search is completed in three stages. First, an initial limited search was conducted in MEDLINE using the terms "early intervention" and "mood disorder" or "anxiety disorder" or "eating disorder" or "personality disorder" or "impulse control disorder" or "substance use disorder" or "somatoform disorder". The initial limited search was conducted by KR in April 2019 to identify keywords and subject headings to generate a search strategy. Different combinations of keywords and subject headings were trialled in MEDLINE, and key papers from the early intervention field were used as indicators for the sensitivity of the search strategy. The preliminary search strategy was developed by KR and reviewed by AA, KA and US. An iterative process was used to balance the sensitivity and specificity. The MEDLINE-specific search strategy returns 3545 documents before deduplication and is outlined in table 1.

In the second stage, all databases will be searched using the MEDLINE search strategy. The search strategy will be tailored to each database. The search for scoping reviews are more iterative than systematic reviews, it is; therefore, feasible that as the reviewers become more familiar with the literature that additional search terms and sources may be identified. The final stage involves identifying additional articles by searching the reference lists of included articles. Studies not reported in English, German, French and Spanish will be excluded from the review during the screening and eligibility assessment. No date limits will be applied to the search. References will be imported to the EndNote X8 reference manager.

### Study selection process

The title and abstract screening in the second stage of the search will be completed by one reviewer with a portion of the articles being screened in duplicate to ensure reliability (25%). Retrieved full texts will also be screened by one reviewer with a sample of full-text documents (25%)

being screened in duplicate for reliability. The eligibility criteria will be applied to each document on a case-by-case basis to determine eligibility for inclusion. Discrepancies between reviewers will be resolved by discussion and if necessary other members of the review team will be consulted.

### Data items and charting

A standardised data charting form developed by the study team will be used to chart the data from eligible studies (see table 2 for a description of each data item). The data charting form was developed using the template from the JBI manual and by drawing on recent reviews of youth service models.[50 51] Each section of the data charting form was developed to address one of the four research questions. The 'Document Details' section which provides descriptive information on document type, author(s), publication date, title and aim/purpose of document will be used to evaluate the extent, nature and range of the literature on early intervention services (question 1). The second section 'Characteristics of Early Intervention Service' will address the second question as key characteristics of the services, namely the population, setting, structure and interventions used in early intervention services will be charted (question 2). The 'Outcome Research' section will be used to answer questions 3 and 4 as any data related to implementation, effectiveness or efficacy will be charted (question 3 and 4). Similar to the full-text screening, one reviewer will chart the majority of the documents with only a portion (25%) of the documents being charted in duplicate to ensure reliability. A small selection of documents will be charted by both reviewers at the outset to ensure that there is clarity and consistency in the use of the data charting form. Where there is more than one paper on the same service model, information will be pooled across the papers to provide the most detailed description of the model and any available evidence.

### Critical appraisal

The lack of critical appraisal tools in scoping reviews has been highlighted as one of the primary limitations of this knowledge synthesis method.[60] Critical appraisal can facilitate the interpretation of reviews by identifying the relative strengths and weaknesses of the included articles and identifying gaps in the research field. However, formal evaluations of methodological quality for scoping reviews can be challenging given the diversity of study designs and the volume of included literature.[61] Given the range of study designs, a two-stage assessment of methodological quality will be conducted for this review. First, each study will be ranked using the JBI Levels of Evidence for Effectiveness from high (level 1) to low (level 5) (level 1—Experimental Designs; level 2—Quasi-experimental Designs; level 3—Observational–Analytical; level 4—Observational–Descriptive; level 5—Expert Opinion and Bench Research).[62] Once stratified according to the level of evidence, the quality of the studies within each stratum

will be evaluated using the JBI Critical Appraisal tools.[63] Additionally, the generalisability and real-world applicability (external validity) of the included studies will be evaluated against the domains of the reach, effectiveness, adoption, implementation and maintenance (RE-AIM) framework. A modified version of a RE-AIM framework rating system developed by Gaglio *et al* will be used in the current study.[64] The modified rating system can be seen in table 3. Each document will be given a rating ranging from 1 (limited generalisability or no information) to 3 (generalisable/pragmatic or information to enable generalisation) on six key domains: participant representativeness, setting representativeness, outcome representativeness, fidelity/adaptation, cost/feasibility of intervention and sustainment. A narrative summary of the methodological quality will be provided alongside quantitative values for each domain of the RE-AIM framework. A portion of the included articles will be appraised in duplicate.

## Synthesis of results

The search results will be reported using a flow diagram to clearly detail the review decision process, indicating the number of citations screened, duplicates removed, study selection and full texts retrieved. The characteristics of the included studies will be presented in an informative table with a narrative and quantitative (eg, frequencies) summary in text. Figures will be used to display the distribution of documents over time and across diagnoses. Descriptions of the early intervention services will be reported for each diagnostic group and transdiagnostically along with any evidence supporting the services and barriers and facilitators to implementation. An aggregated summary of early intervention services with descriptions of common themes and differences across the services will be provided. An effort will be made to identify gaps in knowledge to inform the direction of future research.

## Patient and public involvement

No patients or public were involved in the development of this protocol.

## DISSEMINATION

This review contributes to the growing body of research for early intervention initiatives in mental health by mapping the existing literature on early intervention services for non-psychotic mental health disorders. Through the publication of the results and dissemination via social media and conference presentations, the results will hopefully provide a timely foundation for cross-disciplinary discourse and early intervention service development and research. The results of this review may inform the design of new services and policies to support them.

**Contributors** All authors contributed to the development of this protocol. KR drafted the manuscript and search strategy. AA, KA and US reviewed the search strategy and the draft manuscripts. KR incorporated the feedback from the authors. All authors read and approved the final manuscript.

**Funding** KR and AA are supported by the Health Foundation Scaling Up programme. US is supported by a National Institute of Health Research (NIHR) Senior Investigator Award and receives salary support from the NIHR Mental Health Biomedical Research Centre at the South London and Maudsley NHS Foundation Trust and King's College London. KA is supported by the NHS Innovation Accelerator programme.

**Competing interests** None declared.

**Patient consent for publication** Not required.

**Ethics approval** The synthesis of existing knowledge will not require ethical approval.

**Provenance and peer review** Not commissioned; externally peer reviewed.

**ORCID iDs**
Katie Richards http://orcid.org/0000-0003-3826-6317
Amelia Austin http://orcid.org/0000-0002-4979-4847

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
