## [Reviewer comments · BMJ Open]

ARTICLE DETAILS

TITLE (PROVISIONAL)	Early intervention services for non-psychotic mental health disorders: a scoping review protocol
AUTHORS	Richards, Katie; Austin, Amelia; Allen, Karina; Schmidt, Ulrike

VERSION 1 – REVIEW

REVIEWER	Frank Iorfino Brain and Mind Centre, the University of Sydney
REVIEW RETURNED	05-Sep-2019

GENERAL COMMENTS	The authors present a scoping review protocol on a timely and important topic in mental health – a transdiagnostic early intervention approach to mental health services (namely, beyond psychosis). This work is worthy of publication, provided the comments below are appropriately addressed, specifically the gaps in background that are crucial for this area of research. The introduction for this protocol leaves out a considerable proportion of the available evidence in this field, namely early intervention for mental disorders (beyond psychosis). Many of the key references in the introduction refer to the psychosis literature, however there are a number of key reviews and viewpoints from the non-psychosis literature that is missing; namely those published by Hickie et al, J.Scott et al. I would recommend that the authors enhance the theoretical support for conducting this review by including some of these key references. The levels of evidence presented in the critical appraisal section need to include detail about how the authors plan to evaluate real-world applicability. While, the levels of evidence presented are a common and accepted view, it is important to recognize that experimental designs are not always best suited for health services research given their lack of generalisability due to highly restricted sampling that typically don't materialise in the real world. This is a major barrier to assessing the impact of early intervention services and needs to be considered. Can the authors please comment and make note in the manuscript about how they intend to evaluate evidence based on their generalisability and clinical validity? The authors appropriately argue against a 'siloes' approach to services in the introduction, however in the synthesis of results section, they resort back to a diagnostic approach. While, this is important, it would be good for the authors to mention how they plan to evaluate the utility of transdiagnostic early intervention approaches that aren't limited to specific diagnostic categories. The authors mention clinical staging in the introduction, do the
---

	authors intend to review clinical staging studies in this area given their role in the early intervention field for youth mental health? The title for this paper may need to be amended. Given the paper excludes psychosis (an important part of the early intervention piece), the authors may need to be more explicit that this paper focuses on disorders not including psychosis.
--	--

REVIEWER	Toula Kourgiantakis University of Toronto Canada
REVIEW RETURNED	08-Sep-2019

GENERAL COMMENTS	Abstract Your abstract refers to following the JBI, but does not mention the PRISMA-ScR which is referenced in the protocol. This is also noted in the strengths and limitations. The PRISMA-ScR is an important guide and should be referenced in both places. This sentence in your abstract needs to be re-worded "Data analysis will primarily involve a qualitatively summary of the included articles exploring the characteristics of early intervention services and barriers and facilitators to implementation." Introduction The sentence below needs to be re-worded with a more suitable description of the environment than the word ruthless. "An ever-growing population accompanied by reducing health budgets, creates a ruthless environment where only services that demonstrate effectiveness, economic viability and sustainability receive funding." [38] Method If you are using the JBI framework, did you register your protocol? You need to provide the registration date and number. Your method states that you used the PRISMA-ScR framework, but did you adhere to the scoping review checklist? A copy should be appended to your protocol. Eligibility criteria It would be helpful for the reader if your inclusion/exclusion criteria were numbered. The inclusion criteria state that you will include documents, but I recommend writing more explicitly if you are including empirical literature, grey literature, conceptual/theoretical papers, and reviews. Your last sentence in that paragraph notes that you will include different types of studies which leads the reader to believe that you may be including empirical studies only, but this is unclear. You also note that you will accept mixed child and adolescent literature, but it would be important to consider what you will do if there are studies where it is not possible to distinguish which part of the data is adolescent versus child results. Are you accepting studies examining dual diagnoses or concurrent disorders? Is a diagnosis necessary or could your review include studies examining early intervention for mental health concerns that do not have a diagnosis? Search strategy I can see that you are mentioning grey literature in this section. It is confusing and should be mentioned in your inclusion/exclusion criteria and also explain the other types of articles/studies that are eligible. It is important to explain how this search strategy was developed. How will you identify if this is an early intervention service? Charting It is important to explain why you have chosen your categories and
---

	provide more detail on the category definitions. Diagnosis is an important charting category to include in your charting form. The category "service characteristics" needs a description of what you are trying to extract. You may need to include categories that permit you to chart information related to the early intervention (e.g. when is service offered after diagnosis). It would be important to explain that the 25% of charting conducted by a second team member will be tested at the outset to ensure that there is consistency in the definitions and charting methods used. Overall your topic is relevant and an interesting contribution to the literature on mental health treatment and services. There are a few areas that need to be clarified to strengthen this scoping review.
--	--

VERSION 1 – AUTHOR RESPONSE

Reviewer: 1

1. Reviewer comment: The introduction for this protocol leaves out a considerable proportion of the available evidence in this field, namely early intervention for mental disorders (beyond psychosis). Many of the key references in the introduction refer to the psychosis literature, however there are a number of key reviews and viewpoints from the non-psychosis literature that is missing; namely those published by Hickie et al, J.Scott et al. I would recommend that the authors enhance the theoretical support for conducting this review by including some of these key references.

Authors' response: We appreciate the reviewer's comment and agree that a greater emphasis on the non-psychotic literature would strengthen the rationale for the manuscript. We have attempted to broaden the focus of the introduction by including information on transdiagnostic clinical staging models and early intervention approaches from work published by Hickie and Scott. We have incorporated ideas from Cross & Hickie (2017), Scott et al. (2013), Iorfino et al. (2019), and Cross et al. (2014). The third and fourth paragraphs of the introduction on pages 3 and 4 have been updated to include these details.

2. Reviewer comment: The levels of evidence presented in the critical appraisal section need to include detail about how the authors plan to evaluate real-world applicability. While, the levels of evidence presented is a common and accepted view, it is important to recognize that experimental designs are not always best suited for health services research given their lack of generalisability due to highly restricted sampling that typically don't materialise in the real world. This is a major barrier to assessing the impact of early intervention services and needs to be considered. Can the authors please comment and make note in the manuscript about how they intend to evaluate evidence based on their generalisability and clinical validity?

Authors' response: The Joanna Briggs Institute (JBI) levels of evidence and critical appraisal tools are a useful method for evaluating the internal validity of studies, specifically the clarity of the research question, the study design, and the quality of data acquisition and analysis (e.g. appropriate statistical tests). However, hierarchies of evidence can be criticised as they automatically promote experimental studies above other study designs. We agree that evaluating the real-world applicability of the evidence and the interventions is of paramount importance. The RE-AIM (Reach, Effectiveness, Adoption, Implementation, and Maintenance) framework will be applied to studies to evaluate the generalisability and applicability of the findings. The RE-AIM framework was specifically developed to address the issue of translating research into practice and helps refocus the evaluation of validity from internal to external. We will draw on the methods outlined by Gaglio, Phillips, Heurtin-Roberts, Sanchez and Glasgow (2014). The authors used a 1 (closer to an explanatory trial) to 5 (closer to a pragmatic trial) scale to rate the following items: Participant Representativeness, Setting

Representativeness, Fidelity/Adaptation, Sustainment, and Cost/Feasibility of Treatment. We will be using a modified version of their original criteria. Instead of each item being rated for being closer to a pragmatic versus explanatory trial, we will use a smaller rating scale (1 to 3) and focus on rating the evidence for generalisability (1 = limited information and/or generalisability to 3 = generalisable). We have also included an additional descriptive anchor between 1 and 3 to enable more consistent ratings. Information about this critical appraisal tool have been added to the critical appraisal section on page 15-16. We have also included a table on pages 17-18 of the manuscript which gives a detailed description of the domains and the rating system. Note that this rating system will only be appropriate for interventional trials of early intervention services and will not be applied to purely theoretical papers or position statements that may only include descriptions of services.

3. Reviewer comment: The authors appropriately argue against a 'siloed' approach to services in the introduction, however in the synthesis of results section, they resort back to a diagnostic approach. While, this is important, it would be good for the authors to mention how they plan to evaluate the utility of transdiagnostic early intervention approaches that aren't limited to specific diagnostic categories.

Authors' response: Evaluating the utility of a diagnostic and transdiagnostic approach to early intervention is an important aim of this review. We are particularly interested in looking at differences and similarities across different types of services, including those that are transdiagnostic. We have slightly modified the wording in the synthesis of results section on page 16 to more accurately reflect this.

4. Reviewer comment: The authors mention clinical staging in the introduction, do the authors intend to review clinical staging studies in this area given their role in the early intervention field for youth mental health?

Authors' response: We agree that clinical staging plays an important role in the early intervention field. However, an explicit review of clinical staging studies seems beyond the scope of this review. The focus of the review will be on identifying the range of early intervention literature, characteristics of service models, and key aspects related to their evaluation and implementation. We are particularly interested in mapping the extent, range, and nature of the research in this area. However, considering the reviewer's comment and the central role played by clinical staging, we have now amended the charting form to include a section on clinical staging. This section of the form involves identifying whether the early intervention service was explicitly informed by a clinical staging approach (please see page 14).

5. Reviewer comment: The title for this paper may need to be amended. Given the paper excludes psychosis (an important part of the early intervention piece), the authors may need to be more explicit that this paper focuses on disorders not including psychosis.

Authors' response: We have changed the title to "Early intervention services for non-psychotic mental health disorders: a scoping review protocol" to more explicitly highlight that the paper focuses on disorders other than psychosis.

Reviewer: 2

1. Reviewer comment: Your abstract refers to following the Joanna Briggs Institute (JBI), but does not mention the PRISMA-ScR which is referenced in the protocol. This is also noted in the strengths and limitations. The PRISMA-ScR is an important guide and should be referenced in both places.

Authors' response: We agree that the PRISMA-ScR is an important guide and both the abstract and strengths and limitations sections have been amended to refer to this checklist (please see pages 1 and 2).

2. Reviewer comment: This sentence in your abstract needs to be re-worded “Data analysis will primarily involve a qualitatively summary of the included articles exploring the characteristics of early intervention services and barriers and facilitators to implementation.”

Authors’ response: We appreciate the reviewer highlighting the need to change the wording of this sentence. The sentence has been modified to the following “Data analysis will primarily involve a qualitatively summary of the early intervention literature, the characteristics of early intervention services, and key findings relating to their evaluation and implementation.” (please see page 1).

3. Reviewer comment: The sentence below needs to be re-worded with a more suitable description of the environment than the word ruthless. "An ever-growing population accompanied by reducing health budgets, creates a ruthless environment where only services that demonstrate effectiveness, economic viability and sustainability receive funding." [38]

Author response: We have now omitted the term ‘ruthless’ from the text (please see paragraph 5 of the introduction on pages 4/5).

4. Reviewer comment: If you are using the JBI framework, did you register your protocol? You need to provide the registration date and number.

Authors’ response: We are very much in agreement with the need to register protocols a priori. The title of the review has been registered on the JBI website (https://joannabriggs.org/research/registered_titles.aspx). In line with the aims of pre-registration, the rationale for publishing this protocol in BMJ Open was to increase the transparency of the review process and prevent any unnecessary duplication.

5. Reviewer comment: Your method states that you used the PRISMA-ScR framework, but did you adhere to the scoping review checklist? A copy should be appended to your protocol.

Authors’ response: The PRISMA extension for scoping reviews checklist was used to inform the development of the protocol. We have inserted the term ‘checklist’ into the following sentence “The PRISMA extension for scoping reviews (PRISMA-ScR) checklist [41] and the scoping review framework outlined in the Joanna Briggs Institute Reviewer’s Manual [43]. . . .” (page 5) to provide further clarity that the checklist was used to support protocol development. A copy of the checklist has been included in the table on pages 5 to 8.

6. Reviewer comment: Eligibility criteria: (1) It would be helpful for the reader if your inclusion/exclusion criteria were numbered. (2) The inclusion criteria state that you will include documents, but I recommend writing more explicitly if you are including empirical literature, grey literature, conceptual/theoretical papers, and reviews. Your last sentence in that paragraph notes that you will include different types of studies which leads the reader to believe that you may be including empirical studies only, but this is unclear. (3) You also note that you will accept mixed child and adolescent literature, but it would be important to consider what you will do if there are studies where it is not possible to distinguish which part of the data is adolescent versus child results. (4) Are you accepting studies examining dual diagnoses or concurrent disorders? (5) Is a diagnosis necessary or could your review include studies examining early intervention for mental health concerns that do not have a diagnosis?

Authors response: We very much appreciate this comprehensive feedback on the eligibility criteria. We have addressed each of these points in turn.

- (1) The format of the eligibility criteria section has been modified so that the inclusion/exclusion criteria are now numbered (page 9).
- (2) The description for the inclusion criteria for study and document type has been extended to further highlight that all study and document types are eligible for inclusion (see fourth inclusion criterion in the eligibility section on page 9).
- (3) We are aiming to be as inclusive as possible within this scoping review as early intervention services for youth mental health are an emerging and diverse field. We will, therefore, include literature from mixed child and adolescent services, even if the components for each age group cannot be separated (the third inclusion criterion in the eligibility section on page 9 has been modified to more accurately reflect this).
- (4) Early intervention services examining comorbid/concurrent disorders will be included and this is detailed in the second inclusion criterion on page 9.
- (5) We will be reviewing papers that evaluate both threshold and subthreshold disorders. The term subthreshold has been included in the manuscript to clarify that this is the case on page 9.

7. Reviewer comment: Search strategy: (1) I can see that you are mentioning grey literature in this section. It is confusing and should be mentioned in your inclusion/exclusion criteria and also explain the other types of articles/studies that are eligible. (2) It is important to explain how this search strategy was developed. (3) How will you identify if this is an early intervention service?

Authors response: Thank you, like before we have tried to address each point in turn. We have now modified the search strategy as outlined in the response to reviewer 2, comment 7.

- (1) We have expanded on the description of the types of studies and documents that will be included in the eligibility section on page 9. Grey literature has been added to the inclusion criteria.
- (2) An initial limited search was conducted to generate preliminary keywords and subject headings. The search strategy was iteratively developed by trialling different combinations of keywords and subject headings in MEDLINE. Key research papers within the early intervention literature were used as an indicator of the sensitivity of the search strategy. There was a focus on balancing sensitivity and specificity. The final search strategy included in the initial protocol was less specific and broader than was ideal, it returned many irrelevant papers. However, we kept this broader search strategy to ensure that all the key papers were returned in the MEDLINE search. Since submitting the protocol for review, we have run this search strategy in the other databases. Once we began running the search strategy in the other databases it became apparent that we were getting too many irrelevant papers. We then trialled a simpler search strategy in this broader set of databases. Using a simpler strategy but in a broader set of databases we were able to obtain all the key papers. Given this, we have now modified the search strategy to be simpler, replacing “exp Early Medical Intervention [MeSH term]/ or ((early adj1 (intervent* or treat* or recogni* or detect* or service*)) or ((first or initial) adj1 (admission* or hospital* or episode*)) or (early-intervention* AND service*) or (early intervention* AND service*)).tw.” with “exp Early Medical Intervention [MeSH term]/ or (early intervention* or early-intervention*).tw”. The simplification of the search strategy will make the search more feasible. Please see the amended search strategy and results on pages 10 to 11. Further details of the search strategy development have been added to the search strategy section on page 9-10.
- (3) The identification of whether a paper is describing an early intervention service will be done on a case-by-case basis during the screening stage outlined in the “study selection process” section on page 11. It is difficult to have a very precise and prescriptive description regarding early intervention services. The format, content and structure of such services are diverse. For the current review, early intervention refers to a structured programme of care delivered by a stand-alone team or teams integrated into mental health services that provide treatment for individuals with recent-onset subthreshold or threshold disorders. The level of care can vary from low-intensity techniques of signposting, psychoeducation, and self-help resources all the way through to specialised multi-disciplinary teams and complex high intensity interventions. The first eligibility criterion on page 9 has

been updated with the abovementioned definition to provide further clarity on how we would determine whether a paper is describing an early intervention service.

8. Reviewer comment: Charting: (1) It is important to explain why you have chosen your categories and provide more detail on the category definitions. (2) Diagnosis is an important charting category to include in your charting form. (3) The category "service characteristics" needs a description of what you are trying to extract. (4) You may need to include categories that permit you to chart information related to the early intervention (e.g. when is service offered after diagnosis). (5) It would be important to explain that the 25% of charting conducted by a second team member will be tested at the outset to ensure that there is consistency in the definitions and charting methods used.

Authors' response: Again, we very much appreciate the suggestions and comments in relation to the methods of the review.

(1 & 3) We have attempted to improve the clarity of the data charting form by being more explicit about what will be included in each item and breaking down the larger items such as "service characteristics" into smaller more precise terms. We have also tried to link the items in the form more clearly to our research questions so that readers may understand why we have chosen these items. The data charting form was developed from the template provided in the JBI manual and by reviewing related scoping reviews and protocols. These details have been included in the section 'Data items and charting' on page 11-12 and in the data charting form on pages 12-15.

(2) Diagnosis has been included in the charting form under the headings Population and Participants. We hope that the increased level detail we have provided for category definitions improves the readers ability to understand what details are charted under each item.

(4) Information relating to the early intervention aspect of the service will be charted under population (e.g. individuals between 16-30 years old within 3 years of onset) and service structure/process (e.g. wait time targets of 2 weeks for assessment).

(5) We agree that it is important to test out the data charting form to ensure consistency in definition and method. The following sentence has been added to the data charting section on page 12 to clarify this: "A small selection of documents will be charted by both reviewers at the outset to ensure that there is clarity and consistency in the use of the data charting form."

References

Cross SP, Hermens DF, Scott EM, et al. A clinical staging model for early intervention youth mental health services. *Psychiatr Serv* 2014;65(7):939-43. doi: 10.1176/appi.ps.201300221 [published Online First: 2014/05/16]

Cross SP, Hickie I. Transdiagnostic stepped care in mental health. *Public Health Res Pract* 2017;27(2):1. doi: 10.17061/phrp2721712

Gaglio, B., Phillips, S. M., Heurtin-Roberts, S., Sanchez, M. A., & Glasgow, R. E. (2014). How pragmatic is it? Lessons learned using PRECIS and RE-AIM for determining pragmatic characteristics of research. *Implementation Science*, 9(1), 96.

Iorfino F, Scott EM, Carpenter JS, et al. Clinical stage transitions in persons aged 12 to 25 years presenting to early intervention mental health services with anxiety, mood, and psychotic disorders. *JAMA Psychiatry* 2019 doi: 10.1001/jamapsychiatry.2019.2360

Scott J, Leboyer M, Hickie I, et al. Clinical staging in psychiatry: a cross-cutting model of diagnosis with heuristic and practical value. *The British journal of psychiatry: the journal of mental science* 2018;202(4):243-45. doi: 10.1192/bjp.bp.112.110858 [published Online First: 01/02]